# The effect of access to water, sanitation and handwashing facilities on child growth indicators: Evidence from the Ethiopia Demographic and Health Survey 2016

**Tolesa Bekele** [1,2]*, **Bayzidur Rahman**[1], **Patrick Rawstorne**[1]

1 School of Public Health and Community Medicine, University of New South Wales, Sydney, Australia,
2 Department of Public Health, College of Medicine and Health Sciences, Ambo University, Oromia, Ethiopia

* t.okuba@unsw.edu.au

## Abstract

**Data Availability Statement:** This study has used an existing datasets in the DHS Program data repository that are freely available online with non-identifiable information (https://www.idhsdata.org).

### Introduction

Poor access to water, sanitation, and handwashing (WASH) facilities frequently contribute to child growth failure. The role of access to WASH facilities on child growth outcomes in Ethiopia is largely unknown. The aim of this study was to determine individual and combined effects of access to WASH facilities on child growth outcomes.

### Methods

Data for this analysis was sourced from the recent Ethiopia Demographic and Health Survey (EDHS) 2016. A multivariable logistic regression model was applied to identify the separate and combined association of access to WASH facilities with child growth outcomes. Odds ratio (OR) and 95% confidence interval (CI) were estimated. Statistical significance was declared at $p < 0.05$.

### Results

Included in the analyses were data for children 0–59 months of age, which amounted to valid data for 9588 children with a height-for-age z-score (HAZ), 9752 children with a weight-for-age z-score (WAZ) and 9607 children with a weight-for-height z-score (WHZ). Children with access to improved combined sanitation with handwashing facilities had 29% lower odds of linear growth failure (stunting) (adjusted odds ratio (AOR) = 0.71; 95% CI: 0.51–0.99) compared with those with unimproved. Children with access to combined improved WASH facilities were 33% less likely to have linear growth failure (AOR = 0.67; 95% CI: 0.45–0.98). Access to improved handwashing alone reduced the odds of being underweight by 17% (AOR = 0.83; 95% CI: 0.71–0.98) compared with unimproved. Improved water and sanitation separately as well as combined WASH were not associated with decreased odds of underweight and wasting.

**Funding:** The authors received no specific funding for this work.

**Competing interests:** The authors have declared that no competing interests exist.

## Conclusions

Combined access to improved water, sanitation and handwashing was associated with reduced child linear growth failure. Further research with robust methods is needed to examine whether combined WASH practices have synergistic effect on child growth outcomes.

## Introduction

Child growth failure (CGF), which refers to under-5 stunting, wasting and being underweight, is a specific subset of child under-nutrition that excludes micronutrient deficiencies [1]. Estimations of stunting, wasting and being underweight can serve as a comprehensive assessment of CGF [2]. A child is stunted, underweight or wasted if the z-score for height-for-age, weight-for-height, or weight-for-age, respectively is greater than -2 SDs below the World Health Organization (WHO) median for the healthy refence population [3, 4].

According to a 2019 United Nations International Children's Emergency Fund (UNICEF) report, an estimated 21.9% and 7.3% of under-5 children globally were stunted and wasted, respectively [5]. This burden is concentrated in low- and middle-income countries, where almost all stunted, wasted or underweight children live [5, 6]. Asia and Africa are the two most disproportionately affected CGF regions in the world. According to the joint estimates of UNICEF and its collaborators, in 2018, nearly 55% of the children in Asia were estimated to be stunted and 68% wasted whereas, in Africa, 39% of the children were estimated to be stunted and 28% wasted [5].

Safe drinking water, effective sanitation and adequate hygiene are together referred to as WASH, and have been shown to be key drivers of human health, nutrition, education and gender equality [7]. In 2015, an estimated 663 million of the world's population did not have access to improved drinking water sources, of which half of these people were in Sub-Saharan Africa [8]. While nearly 2.4 billion or 1/3$^{rd}$ of the world's population lacked access to improved sanitation facilities and 13% practiced open defecation [9]. Although, it is difficult to obtain reliable worldwide estimates on hygiene practices through handwashing, Freeman and colleagues in 2014 estimated that only 19% of the world's population washed their hands after contact with excreta and only 14% of people in Sub-Saharan Africa wash their hands with soap after defecation and before eating [10]. Insufficient food intake, repeated infections, deficiencies in micronutrient, and the dearth of access to WASH all contribute to child growth failure.

WASH offers a possible solution to CGF in many countries, and its global importance is recognized in the Sustainable Development Goal (SDG) 6 while CGF is also acknowledge in SDG 2.2 [11]. WASH may reduce CGF in three ways: by reducing the incidence of diarrheal disease [12]; by preventing intestinal worm infection that contributes to inadequate absorption of nutrients [13] and by reducing the pathogen load in the environments as a result of poor WASH condition [13–15]. Despite the potential benefits of WASH, few studies have investigated the association of inadequate WASH with CGF compared with diarrhea and soil-transmitted helminth infections [9]. The paucity of evidence on the effect of WASH services on child growth is argued to be as a result of relatively low priority given to WASH research among medical researchers [16]. However, recently there is a great need for this type of research as CGF in many Sub-Saharan Africa countries such as Ethiopia remains at unacceptably high levels suggesting that WASH may be a solution.

In Ethiopia, CGF has decreased only marginally over the past decade, and remains at high levels, with an estimated 38% of children stunted, 24% underweight and 10% wasted [17].

Children in rural areas are more likely to be malnourished compared with children in urban areas, with variations in the prevalence of stunting and wasting by region [17]. Although Ethiopia did not achieve the Millennium Development Goal 7, which included access to basic sanitation as well as clean and safe drinking water, it made reasonable progress towards the goal. Nonetheless, when the MDG period ended in 2015, nearly 49% of the rural population and 39% of the urban population remained without access to improved sanitation, and 50% of the rural population were still using unimproved drinking water [18]. According to the 2016 EDHS, in Ethiopia nearly 34 million people (most of them in the rural areas) were reported to be without access to a safe water supply and nearly 48 million had no access to basic sanitation [17].

Strategies have been established in Ethiopia to tackle child growth failure via nutrition interventions and diarrheal disease control [19]. However, the extent to which inadequate access and practices of WASH have been contributing to child growth failure remains inadequate understood. Challenges to interpret and synthesis study findings due to variations in methods used and study areas are limiting the provision of potential policy recommendations [20]. In addition, complex interaction among WASH components may also pose differential effect on child growth outcomes [9]. This study seeks to determine the role of household access to WASH facilities separately or when combined on child growth outcomes in Ethiopia. The study assesses how WASH factors are associated with stunting, underweight and wasting among children under-5 years old. It is anticipated the current findings will help to guide policy and programs designed to deliver targeted intervention strategies and to identify entry points for child growth failure.

## Methods

### Data and population

Data used in this analysis were obtained from the 2016 Ethiopia Demographic and Health Survey in the children's file [21]. The Ethiopian DHS was financially sponsored by the United States Agency for International Development (USAID). The survey was implemented in collaboration with the Ethiopian Ministry of Health, Central Statistical Agency (CSA) and ICF International (previous Macro International). Population and health indicators were collected from all 9 regional states and 2 cities administrates. In 2007, the population and housing census (PHC) was conducted by the CSA and provided a sampling frame from which the EDHS 2016 samples were drawn. The source population included all children aged 0–59 months as well as their mothers or caregivers in the enumeration areas of the EDHS who slept in the selected households the night before the survey. Included in the analyses were all children under-5 years of age and women aged 15–49 years whose height and weight measurements were taken in all selected households.

### Data access, sample size and sampling

The 2016 EDHS data were downloaded from the DHS Program website using the Integrated Public Use Microdata Series extract system [21]. The 2016 EDHS used a two-stage stratified cluster sampling method to identify a nationally representative sample of the Ethiopian population living in 9 regions and 2 city administrations of the country. The census sampling frame was a complete list of 84, 915 enumeration areas (EAs) created for the 2007 PHC. An EA is a geographic area consisting of on average 181 households. In the first sampling stage, a total of 645 EAs or clusters (202 in urban and 443 in rural areas) were randomly selected with probability proportional to EA size (based on the 2007 PHC) and with independent selection in each sampling stratum. In the second stage, a fixed number of 28 households per EA or cluster were

selected with an equal probability of selection from the household list. A total of 18008 house-holds were selected for the 2016 EDHS, of which participants from 16650 (92.5%) households provided data. The study participants for the current analyses included 10641 unweighted child-mother or caregiver pairs.

## Data collection

The 2016 EDHS used a questionnaire that was adapted from model survey tools developed for The DHS Program project. The questionnaire was written in English and then translated into three local languages: *Afan Oromo*, *Amharic and Tigrigna* by language experts. All information related to children and mothers or caregivers were taken at home by interviewing mothers or caregivers. WASH indicators were also collected through face to face interviews and observation methods. The 2016 EDHS research team took measurements of height and weight for all children under five years as well as for their mothers or caregivers. In the EDHS blood samples were drawn from a drop of blood taken from a finger prick women and children (or a heel prick in the case of children age 6–59 months) and collected in a microcuvette. Haemoglobin analysis was performed on-site using a battery-operated portable HemoCue analyser. Details on the survey sampling procedure and data collection methods are described elsewhere [17].

## Measures

Since the DHS sampling design included both under- and over-sampling, all analyses were adjusted through weighting. Child growth outcomes (i.e. study outcome) were shown by three indicators: stunting, underweight and wasting using the WHO 2006 Child Growth Standard [3, 4]. Child linear growth failure is known as stunting (i.e. an abnormal slow rate of gain in child's height or length) [22] which indicates chronic undernutrition in children [23]. The main exposure (study factor) variables were WASH indicators categorized as binary variable 0 = improved and 1 = unimproved according to the Joint Monitoring Program [7]. House-holds with access to a river, stream, pond, unprotected spring or well, lake, canal, dam or irrigation channel as the main drinking water sources and which did not also report water treatment using at least one of the methods including boiling, using bleach or chlorine, water filter, solar disinfection, letting water stand and settle, were categorized as having unimproved drinking water sources. An unimproved sanitation facilities included the type of sanitation facility that household members typically used if they reported no facility or the use of the bush or a field, a pit without a slab or an open pit, a bucket toilet, a hanging toilet or an unsewered latrine. Handwashing facility was measured in the EDHS by direct observation. If the observer did not see a specific location for handwashing in the house, yard, or plot during data collection, households were considered to have no handwashing facility and was classified as unimproved handwashing. Minimum food groups and meal frequencies the child had consumed in the 24 hours prior to the survey were constructed for analyses using a the WHO et al [24] classification.

## Statistical analysis

Data management and analysis were performed using Statistical Analysis System (SAS) version 9.4 (SAS Institute Inc., Cary, NC, USA). A univariable logistic regression model (***model 0***) was fitted with each of the explanatory variables to select candidates with p-value < 0.25 for the multivariable (base) model. To compare the effect of each WASH indicator using regression models, we created five different models to reach the final model. By model building process, we first build the best model (*model 1*) which contains explanatory variables that best explain the outcome variable (i.e. all significant variables). We retained significant variables

only in the model (i.e. best model) after dealing with potential confounder and effect modifiers. In the first step, all explanatory variables with p-values < 0.25 from **model 0** (maximum model) were entered the base model without WASH components. Then using step-down procedure, we removed a single term with the highest non-significant p-value at a time until we get the model which contains only significant terms (p <0.05) (**model 1**). In the second step, *water facility* plus significant variables from *model one* were independently modelled with the outcome variable (**model 2**). In the third step, *sanitation facility* and significant variables in *model one* were examined with the outcome variable (**model 3**). In the fourth step, only *handwashing facility* plus significant variables in *model one* were modelled (**model 4**). In the final step, *combined WASH facilities* (water, sanitation and handwashing) and significant variables in *model one* was checked in the final model (**model 5**).

Univariate and multivariable logistic regression models using survey-specific SAS procedures (PROC SURVEYLOGISTIC) were used to examine the association between the risk of child growth failure and access to WASH facilities adjusting for potential confounders. We included potential confounding factors considered to be major immediate (dietary and diseases) and underlying (poverty, inadequate basic services and infrastructure) causes of child growth failure according to the conceptual frameworks of UNICEF [25] and others [26, 27]. Although children 0–5 and 6–59 months of age have different feeding practices, we analyzed data by merging the two age groups because age was not a statistically significant effect modifier. Combined WASH was considered to be present in households with three facilities, including: drinking water, sanitation and handwashing. We fitted multiple liner regression model and invoked the Variance Inflation Factor (VIF) to check for multicollinearity and there was no evidence for multicollinearity (VIF < 2). Odds ratios and corresponding 95% confidence intervals were estimated and statistical significance was set at $p < 0.05$.

## Ethical considerations

Approval to use the 2016 EDHS was sought and received by the DHS Program in the United States of America. Ethical clearance to conduct the EDHS was provided to the relevant research organization by the Ethiopian Public Health Institute, formerly the Ethiopia Health and Nutrition Research Institute Review Board, the National Research Ethics Review Committee at the Ministry of Science and Technology, the Institutional Review Board of ICF International, and Centre for Diseases Control and Prevention in Atlanta. Given that the EDHS data are publicly available upon request and contains non-identifiable data that has been previously collected, the Human Research Ethics Committee at The University of New South Wales (UNSW), Australia, found that the current analyses posed no foreseeable additional risk of harm or discomfort to participants.

## Results

### Study sample characteristics and household WASH facilities

A total of weighted 11023 child-mother or caregiver pairs with complete anthropometric data were included in the analyses. Table 1 shows a weighted frequency distribution of selected characteristics of the study sample. A majority of respondents (88.97%) were of rural origin by place of residence. Nearly two-third of the mothers or caregivers (66.08%) had no formal education while 51.84% of their partners had some formal education. The mean (SD) age of mothers or caregivers was 29.55 (6.73) years and 75.33% had a normal body mass index (BMI) and 69.70% had no anemia. The largest proportion of respondents was found in the lowest wealth quintile (23.92%). The mean (SD) age of the children was 28.67 (18.07) months and 51.94% were male (Table 1).

**Table 1. Socio-demographic and economic characteristics of respondents included in the analysis, 2016 EDHS (n = 11023).**

| Characteristics | Weighted, n (%) |
|---|---|
| **Place of residence** | |
| Urban | 1216 (11.03) |
| Rural | 9807 (88.97) |
| **Region** | |
| Tigray | 716 (6.49) |
| Afar | 114 (1.04) |
| Amhara | 2072 (18.80) |
| Oromia | 4851 (44.01) |
| Somali | 508 (4.61) |
| Benishangul Gumuz | 122 (1.10) |
| SNNP | 2296 (20.83) |
| Gambela | 27 (0.24) |
| Harari | 26 (0.23) |
| Addis Ababa | 244 (2.21) |
| Dire Dawa | 47 (0.43) |
| **Household head** | |
| Male | 9494 (86.13) |
| Female | 1529 (13.87) |
| **Maternal education** | |
| Has education | 3739 (33.92) |
| Has no education | 7284 (66.08) |
| **Maternal age (years)** | |
| Mean (SD) | 29.55 (6.73) |
| Median (IQR) | 28.19 (9.0) |
| **Paternal education** | |
| Has education | 5385 (51.84) |
| Has no education | 5003 (48.16) |
| **Wealth index** | |
| Poorest | 2636 (23.92) |
| Poorer | 2520 (22.86) |
| Middle | 2280 (20.68) |
| Richer | 1999 (18.13) |
| Richest | 1588 (14.41) |
| **Sex of child** | |
| Male | 5725 (51.94) |
| Female | 5298 (48.06) |
| **Age of child (months)** | |
| Mean (SD) | 28.67 (18.07) |
| Median (IQR) | 28.0 (31.0) |

IQR (Interquartile range); SD (Standard deviation); SNNP (Southern Nations, Nationalities and People).

More than half (54.47%) of the children had moderate or mild anemia levels and 88.13% had experienced no diarrhea in the two weeks prior to the survey. About 42.38% of children were of average size at birth. Of the children who were weighed at birth, 13.19% were < 2500 grams at birth. Among children, 5.27% were never breastfed, 52.93% were vaccinated, 21.69% were born within 24 months of a preceding birth, while the mothers of 68.12% had

experienced fewer than four antenatal care (ANC) visits. A small proportion of children (4.14%) in the 24 hours prior to the survey had consumed a minimum diet diversity ($> = 4$ food groups) while 13.63% of children has consumed the meal four to seven times (Table 2).

**Table 2. Child and maternal characteristics from univariable analyses (n = 11023).**

| Child anaemia status | Weighted, n (%) |
|---|---|
| Severe | 262 (3.09) |
| Moderate/mild | 4620 (54.47) |
| Not anaemic | 3600 (42.44) |
| **Childbirth order** | |
| Mean (SD) | 4.01 (2.49) |
| Median (IQR) | 3.07 (4.0) |
| **Size of child at birth (subjective)** | |
| Larger than average | 3485 (31.62) |
| Average | 4672 (42.38) |
| Smaller than average | 2866 (26.00) |
| **Child's birthweight** | |
| < 2500 grams | 198 (13.19) |
| > = 2500 grams | 1304 (86.81) |
| **Breastfeeding status** | |
| Never breastfed | 581 (5.27) |
| Still breastfed | 4813 (43.66) |
| Ever breastfed (not currently) | 5629 (51.07) |
| **Vaccination based on source of information** | |
| Card seen at home/health facility or mothers' report | 3292 (52.93) |
| No card or no longer has card) | 2927 (47.07) |
| **Diarrhoea in past 2 weeks** | |
| Yes | 1227 (11.87) |
| No | 9110 (88.13) |
| **Number of under-5 in the household** | |
| Mean (SD) | 1.80 (0.90) |
| Median (IQR) | 2.0 (1.0) |
| **Maternal BMI (Kg/m$^2$)** | |
| Underweight (< = 18.4) | 1935 (18.05) |
| Normal (18.4 < to 24.9) | 8079 (75.33) |
| Overweight (> 24.9) | 711 (6.63) |
| **Birth interval** | |
| < 24 months | 1942 (21.69) |
| > = 24 months | 7011 (78.31) |
| **Number of ANC visits** | |
| < 4 | 5160 (68.12) |
| > = 4 | 2415 (31.88) |
| **Maternal anaemia status** | |
| Not anaemic | 7416 (69.70) |
| Moderate | 3066 (28.82) |
| Sever | 158 (1.48) |
| **Minimum dietary diversity in 24hrs** | |
| > = 4 food groups | 138 (4.14) |
| < 4 food groups | 3187 (95.86) |

(*Continued*)

**Table 2.** (Continued)

| Child anaemia status | Weighted, n (%) |
|---|---|
| **Minimum meal frequency in 24hrs** | |
| Less than two times | 2032 (49.83) |
| Two to three times | 1490 (36.54) |
| Four to seven times | 556 (13.63) |

IQR (Inter-quartile range); SD (standard deviation).

About 56% of households had access to improved drinking water sources, while 9.94% had access to improved sanitation and 52.85% had access to a handwashing facility with water and soap. Only 6.65% of households had access to combined WASH facilities. The prevalence of stunting, underweight and wasting were 38.39% (95% CI: 36.49–40.29), 23.73% (95% CI: 22.19–25.27) and 10.09% (95% CI: 9.09–11.09), respectively (Table 3).

**Table 3. WASH facilities and prevalence of child growth failure indicators, 2016 EDHS (weighted n = 11023).**

| Characteristics | Weighted, n (%) |
|---|---|
| **Water facility** | |
| Improved | 6220 (56.43) |
| Unimproved | 4803 (43.57) |
| **Sanitation facilities** | |
| Improved | 1095 (9.94) |
| Unimproved | 9928 (90.06) |
| **Handwashing facility** | |
| Improved | 5825 (52.85) |
| Unimproved | 5197 (47.15) |
| **Water + Sanitation** | |
| Improved | 860 (7.80) |
| Unimproved | 10163 (92.20) |
| **Water + handwashing** | |
| Improved | 3615 (32.79) |
| Unimproved | 7408 (67.21) |
| **Sanitation + Handwashing** | |
| Improved | 748 (6.80) |
| Unimproved | 10274 (93.21) |
| **WASH facilities** | |
| Improved | 622 (5.65) |
| Unimproved | 10400 (94.35) |
| **Water collection time** | |
| < = 30 minutes | 5666 (58.08) |
| > 30 minutes | 4089 (41.88) |
| **Water purification done** | |
| Yes | 960 (8.71) |
| No | 10059 (91.25) |
| Do not know | 4 (0.04) |
| **Stunting** * | |
| Stunted | 3681 (38.39) |

(Continued)

**Table 3.** (Continued)

| Characteristics | Weighted, n (%) |
|---|---|
| Normal | 5907 (61.61) |
| **Underweight** * | |
| Underweight | 2314 (23.73) |
| Normal | 7438 (76.27) |
| **Wasting** * | |
| Wasted | 970 (10.09) |
| Normal | 8637 (89.91) |

* Estimated by using the WHO growth reference.

There were regional variations in the proportion of child growth failure. Stunting ranged from a high of 47.17% in the Amhara region to a low of 14.68% in Addis Ababa. The Afar region had the highest proportion of children who were underweight (36.20%) whereas wasting (23.13%) was highest in the Somali region (Fig 1).

## Association between access to WASH facilities and child growth outcomes

**Stunting.** In the analysis of 9588 weighted number of children 0–59 months of age with valid HAZ data (i.e. data for length or height and age), there were 3681 (38.39%) cases of

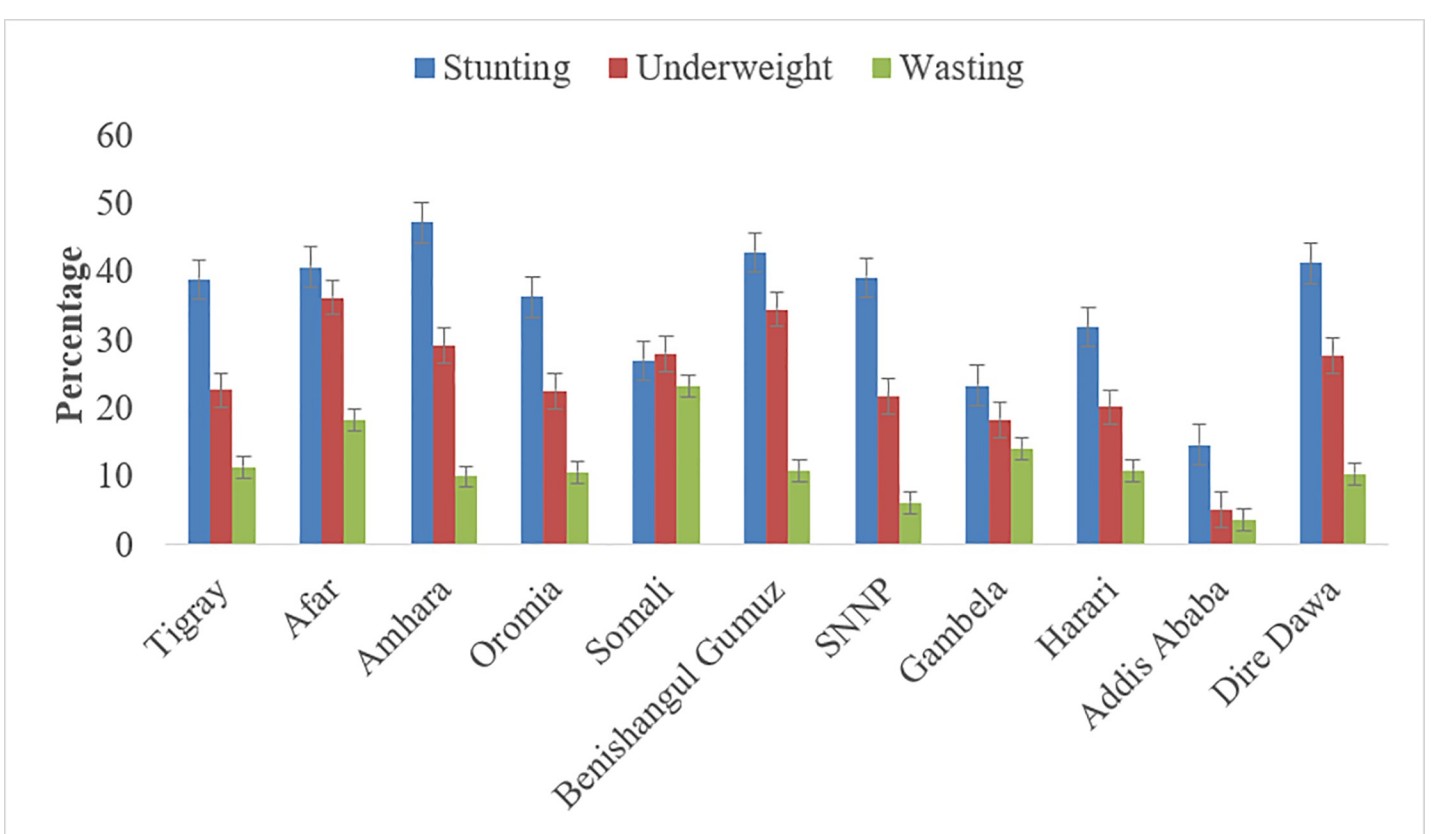

**Fig 1. Regional variation in child growth failure indicators, EDHS 2016.**

stunting. The findings of this study confirmed there was a univariable association between access to combined sanitation with handwashing facilities and stunting (Crude odds ratio (COR) = 0.43; 95% CI: 0.33–0.57, p < 0.001). In a multivariable model, and adjusted for potential confounders, children with access to improved combined sanitation with handwashing facilities had 0.71 times lower odds of being stunted, compared with children with access to unimproved combined sanitation with handwashing facility (AOR = 0.71; 95% CI: 0.51–0.99, p = 0.044). These results are not presented in a table.

Using a model building process, we derived a multivariable model for stunting that comprised region, wealth quintiles, sex of child, child anemia status, child birth order, size of child at birth, breastfeeding status, presence of diarrhea, maternal age, birth interval, number of ANC visits and minimum diet diversity consumed in 24 hours. Table 4 (model 5) shows the association between access to combined WASH facilities and stunting. Compared with children who had access to unimproved WASH facilities, children in improved WASH facilities were 33% less likely to be stunted, adjusting for potential confounders (AOR = 0.67; 95% CI: 0.45–0.98, p = 0.040). Individual access to improved drinking water sources, sanitation and handwashing facilities were significantly associated with stunting in the univariable model but diminished after adjusting for all potential confounders (S1 Table).

The highest odds of being stunted was observed in the Amhara region compared with other regions (AOR = 2.34; 95% CI: 1.48–3.72, p = < 0.001). Children from the highest wealth quintile were 41% less likely to be stunted compared with children from the lowest wealth quintile (AOR = 0.59; 95% CI: 0.44–0.80, p = 0.001). The odds of being stunted were 23% lower among female compared with male children (AOR = 0.77; 95% CI: 0.67–0.89, p = 0.001). The odds of stunting were increased by 43% among children 36 to 47 months of age compared with their younger counterparts (AOR = 7.43; 95% CI: 5.12–10.77, p < 0.001). Children who had no anemia were 55% less likely to be stunted compared with children with severe anemia (AOR = 0.45; 95% CI: 0.29–0.71, p = 0.001). A strong significant association was detected between birth order and stunting, the odds of being stunted were increased by 1.07 for each additional child increase in preceding births (AOR = 1.07; 95% CI: 1.01–1.13, p = 0.021). The size of the child at birth was significantly associated with the risk of being stunted; the odds of stunting were 1.57 times higher (AOR = 1.57; 95% CI: 1.30–1.89, p < 0.001) in children with smaller size than average at birth compared with children larger size than average at birth.

For every one-year increase in the age of the mother between 15 to 49, the odds of being stunted were decreased by 3% adjusting for confounding variables (AOR = 0.97; 95% CI: 0.95–0.99, p = 0.007). The odds of being stunted were 28% higher among children whose mothers had fewer ANC visits (< 4 visits) compared with children whose mothers had four or more ANC visits (AOR = 1.28; 95% CI: 1.08–1.52, p = 0.016). Children who consumed food less than the minimum diet diversity (< 4 food groups) in the 24 hours prior to the survey were 27% more likely to be stunted compared with those who consumed > = 4 food groups (AOR = 2.27; 95% CI: 1.11–4.66, p = 0.028) (Table 4).

**Underweight.** There were 9752 weighted number of children 0–59 months of age in the analysis with WAZ data, of which 2314 (23.73%) were cases of being underweight. In the multivariable model, children with access to improved handwashing facilities had 17% lower odds of being underweight, compared with those who had access to an unimproved handwashing facility alone (water and soap), adjusting for potential confounders (AOR = 0.83; 95% CI: 0.71–0.98, p = 0.032) (S2 Table). Access to improved individual water and sanitation as well as combined WASH facilities were not predictors of being underweight in children after adjusting for confounders.

The odds of a child being underweight were greater in Benishangul-Gumuz (AOR = 4.99; 95% CI: 2.24–11.09, p < 0.001), Amhara (AOR = 3.27; 95% CI: 1.50–7.10, p = 0.003) and Dire

**Table 4. Association between individual and combined access to WASH and stunting among children 0–59 months of age, EDHS 2016 (n = 9588).**

| Variables | Stunting | | Model 0 | | Model 5 | |
|---|---|---|---|---|---|---|
| | No | Yes | COR (95% CI) | p value | AOR (95% CI) | p value |
| **Water facility** | | | | | | |
| Improved | 3402 | 1982 | 0.86 (0.74–0.99) | 0.002 | | |
| Unimproved | 2505 | 1699 | Ref | | | |
| **Sanitation facility** | | | | | | |
| Improved | 684 | 252 | 0.57 (0.44–0.73) | < 0.001 | | |
| Unimproved | 5223 | 3429 | Ref | | | |
| **Handwashing facility** | | | | | | |
| Improved | 3199 | 1906 | 0.91 (0.79–1.04) | 0.040 | | |
| Unimproved | 2707 | 1775 | Ref | | | |
| **WASH facilities** | | | | | | |
| Improved | 420 | 109 | 0.40 (0.29–0.55) | < 0.001 | 0.67 (0.45–0.98) | 0.040 |
| Unimproved | 5477 | 3572 | Ref | | Ref | |
| **Region** | | | | | | |
| Addis Ababa | 180 | 31 | Ref | | Ref | |
| Tigray | 397 | 253 | 3.69 (2.60–5.24) | < 0.001 | 1.76 (1.12–2.78) | 0.015 |
| Afar | 54 | 37 | 3.98 (2.78–5.71) | < 0.001 | 1.13 (0.67–1.92) | 0.639 |
| Amhara | 992 | 886 | 5.19 (3.64–7.40) | < 0.001 | 2.34 (1.48–3.72) | < 0.001 |
| Oromia | 2680 | 1524 | 3.31 (2.33–4.68) | < 0.001 | 1.26 (0.78–2.03) | 0.351 |
| Somali | 290 | 107 | 2.15 (1.49–3.10) | < 0.001 | 0.62 (0.37–1.02) | 0.057 |
| Benishangul-Gumuz | 58 | 43 | 4.36 (3.04–6.25) | < 0.001 | 1.80 (1.11–2.94) | 0.018 |
| SNNP | 1204 | 774 | 3.74 (2.61–5.34) | < 0.001 | 1.59 (1.00–2.53) | 0.051 |
| Gambela | 17 | 5 | 1.77 (1.20–2.61) | 0.004 | 0.75 (0.46–1.21) | 0.237 |
| Harari | 14 | 6 | 2.72 (1.83–4.04) | < 0.001 | 1.27 (0.77–2.10) | 0.355 |
| Dire Dawa | 21 | 15 | 4.08 (2.71–6.14) | < 0.001 | 1.83 (1.19–2.98) | 0.010 |
| **Wealth index** | | | | | | |
| Poorest | 1214 | 997 | Ref | | Ref | |
| Poorer | 1281 | 969 | 0.92 (0.75–1.13) | 0.432 | 0.92 (0.72–1.18) | 0.517 |
| Middle | 1257 | 761 | 0.74 (0.59–0.93) | 0.009 | 0.74 (0.57–0.96) | 0.025 |
| Richer | 1137 | 605 | 0.65 (0.53–0.80) | < 0.001 | 0.69 (0.54–0.87) | 0.002 |
| Richest | 1018 | 349 | 0.42 (0.33–0.53) | < 0.001 | 0.59 (0.44–0.80) | 0.001 |
| **Sex of the child** | | | | | | |
| Male | 2876 | 2017 | Ref | | Ref | |
| Female | 3031 | 1664 | 0.78 (0.69–0.89) | < 0.001 | 0.77 (0.67–0.89) | 0.001 |
| **Age of child (months)** | | | | | | |
| 0–11 | 1755 | 351 | Ref | | Ref | |
| 12–23 | 1127 | 780 | 3.46 (2.65–4.50) | < 0.001 | 4.01 (2.86–5.62) | < 0.001 |
| 24–35 | 927 | 871 | 4.69 (3.66–6.02) | < 0.001 | 6.72 (4.75–9.50) | < 0.001 |
| 36–47 | 972 | 860 | 4.43 (3.50–5.59) | < 0.001 | 7.43 (5.12–10.8) | < 0.001 |
| 48–59 | 1126 | 819 | 3.64 (2.81–4.70) | < 0.001 | 6.92 (4.67–10.2) | < 0.001 |
| **Child anaemia status** | | | | | | |
| Sever | 119 | 133 | Ref | | Ref | |
| Moderate | 2538 | 1991 | 0.70 (0.47–1.05) | 0.08 | 0.76 (0.49–1.18) | 0.221 |
| Not anaemic | 2235 | 1330 | 0.53 (0.36–0.79) | 0.002 | 0.45 (0.29–0.71) | 0.001 |
| **Childbirth order** | 5907 | 3681 | 1.03 (1.00–1.06) | 0.015 | 1.07 (1.01–1.13) | 0.021 |
| **Size of child at birth** | | | | | | |
| Larger than average | 1954 | 1044 | Ref | | Ref | |

*(Continued)*

**Table 4.** (Continued)

| Variables | Stunting | | Model 0 | | Model 5 | |
|---|---|---|---|---|---|---|
| | **No** | **Yes** | **COR (95% CI)** | **p value** | **AOR (95% CI)** | **p value** |
| Average | 2577 | 1526 | 1.11 (0.96–1.28) | 0.151 | 1.15 (0.98–1.35) | 0.078 |
| Smaller than average | 1376 | 1111 | 1.51 (1.27–1.80) | < 0.001 | 1.57 (1.30–1.89) | < 0.001 |
| **Breastfeeding status** | | | | | | |
| Never breastfed | 199 | 146 | Ref | | Ref | |
| Still breastfeeding | 2982 | 1571 | 0.72 (0.50–1.04) | 0.084 | 1.19 (0.78–1.81) | 0.431 |
| Ever breastfed | 2726 | 1964 | 0.98 (0.69–1.41) | 0.930 | 0.76 (0.51–1.13) | 0.178 |
| **Diarrhoea in 2 weeks** | | | | | | |
| Yes | 681 | 470 | 1.12 (0.90–1.39) | 0.143 | 1.15 (0.91–1.46) | 0.348 |
| No | 5210 | 3209 | Ref | | Ref | |
| **Maternal age (years)** | 5907 | 3681 | 1.01 (0.99–1.02) | 0.088 | 0.97 (0.95–0.99) | 0.007 |
| **Birth interval** | | | | | | |
| < 24 months | 901 | 707 | 1.30 (1.09–1.56) | < 0.001 | 1.18 (0.98–1.43) | 0.211 |
| > = 24 months | 3882 | 2341 | Ref | | Ref | |
| **Number of ANC visits** | | | | | | |
| < 4 | 2810 | 1787 | 1.44 (1.23–1.69) | < 0.001 | 1.28 (1.08–1.52) | 0.016 |
| > = 4 | 1498 | 663 | Ref | | Ref | |
| **Minimum food consumed in 24hrs** | | | | | | |
| < 4 food groups | 1845 | 1061 | 2.11 (1.13–3.95) | < 0.001 | 2.27 (1.11–4.66) | 0.028 |
| > = 4 food groups | 95 | 26 | Ref | | Ref | |

AOR (adjusted odds ratio); ANC (antenatal care); COR (crude odds ratio); Ref (reference group); SNNP (southern nations, nationalities and people); WASH (water, sanitation and handwashing); Model 0, results from unadjusted univariable analysis; Model 5, adjusted for combined WASH facilities plus all variables with p value < 0.05 in Model 1.

Dawa (AOR = 3.04, 95% CI: 1.42–6.50, p = 0.004) compared with the capital city, Addis Ababa. Children whose mothers had no formal education were 1.40 times more likely to be underweight compared with children whose mothers had some formal education (AOR = 1.40; 95% CI: 1.15–1.71, p = 0.001). Children from families in a higher wealth quintile (greater wealth) were 44% less likely to be underweight compared with children from families in the lowest wealth quintile (AOR = 0.56; 95% CI: 0.42–0.74, p < 0.001). The odds of being underweight were decreased 19% among female compared with male children (AOR = 0.81; 95% CI: 0.70–0.93, p = 0.004). Children 48 to 59 months of age were 2.97 times (AOR = 2.97; 95% CI: 2.11–4.17, p < 0.001) more likely to be underweight compared with children 0 to 11 months of age. Children who had no anemia had 67% less odds of being underweight compared with children with severe anemia (AOR = 0.33; 95% CI: 0.21–0.53, p < 0.001). Children who were a smaller size than average at birth were more likely to be underweight compared with children who were larger size than average size at birth (AOR = 1.96; 95% CI: 1.58–2.44, p < 0.001). Children with a low birth weight (< 2500 grams) at birth were 2.43 times more likely to be underweight (AOR = 2.43; 95% CI: 1.31–4.52, p = 0.015). The odds of being underweight among children who had diarrhea in the two weeks prior to the study was 1.61 times higher than children who had no diarrhea in that period (AOR = 1.61; 95% CI: 1.26–2.07, p = 0.001). Children of mothers with a BMI > = 18.4 kg/m$^2$ had 51% higher odds of being underweight compared with children of mothers with BMI > 24.9 kg/m$^2$ (AOR = 2.51; 95% CI: 1.67–3.79, p < 0.001) (Table 5).

**Table 5. Association between individual and combined access to WASH and being underweight among children 0–59 months of age, EDHS 2016 (n = 9752).**

| Variables | Underweight | | Model 0 | | Model 5 | |
|---|---|---|---|---|---|---|
| | No | Yes | COR (95% CI) | p value | AOR (95% CI) | p value |
| **Water facility** | | | | | | |
| Improved | 4244 | 1252 | 0.88 (0.74–1.04) | 0.127 | | |
| Unimproved | 3194 | 1062 | Re | | | |
| **Sanitation facility** | | | | | | |
| Improved | 786 | 153 | 0.60 (0.43–0.84) | < 0.001 | | |
| Unimproved | 6652 | 2161 | Ref | | | |
| **Handwashing facility** | | | | | | |
| Improved | 4061 | 1115 | 0.77 (0.67–0.90) | < 0.001 | | |
| Unimproved | 3377 | 1199 | Ref | | | |
| **WASH facilities** | | | | | | |
| Improved | 484 | 61 | 0.39 (0.26–0.61) | < 0.001 | 0.78 (0.48–1.27) | 0.313 |
| Unimproved | 6954 | 2253 | Ref | | Ref | |
| **Region** | | | | | | |
| Addis Ababa | 202 | 11 | Ref | | Ref | |
| Tigray | 509 | 148 | 5.36 (2.77–10.38) | < 0.001 | 2.37 (1.11–5.09) | 0.026 |
| Afar | 60 | 34 | 10.41 (5.30–20.5) | < 0.001 | 2.80 (1.25–6.25) | 0.012 |
| Amhara | 1344 | 552 | 7.53 (3.88–14.64) | < 0.001 | 3.27 (1.50–7.10) | 0.003 |
| Oromia | 3317 | 963 | 5.33 (2.76–10.29) | < 0.001 | 2.23 (1.01–4.91) | 0.047 |
| Somali | 293 | 114 | 7.14 (3.65–13.97) | < 0.001 | 2.35 (1.07–5.16) | 0.033 |
| Benishangul-Gumuz | 68 | 36 | 9.65 (4.85–19.21) | < 0.001 | 4.99 (2.24–11.1) | < 0.001 |
| SNNP | 1584 | 438 | 5.08 (2.60–9.93) | < 0.001 | 2.41 (1.11–5.23) | 0.026 |
| Gambela | 18 | 4 | 4.11 (2.03–8.30) | < 0.001 | 1.87 (0.85–4.14) | 0.122 |
| Harari | 16 | 4 | 4.63 (2.33–9.21) | < 0.001 | 2.38 (1.08–5.24) | 0.031 |
| Dire Dawa | 27 | 10 | 7.01 (3.51–14.02) | < 0.001 | 3.04 (1.42–6.50) | 0.004 |
| **Maternal education** | | | | | | |
| Has education | 2793 | 558 | Ref | | Ref | |
| Has no education | 4645 | 1756 | 1.89 (1.58–2.27) | < 0.001 | 1.40 (1.15–1.71) | 0.001 |
| **Wealth index** | | | | | | |
| Poorest | 1574 | 705 | Ref | | Ref | |
| Poorer | 1658 | 625 | 0.84 (0.69–1.03) | 0.096 | 0.94 (0.74–1.21) | 0.638 |
| Middle | 1562 | 472 | 0.68 (0.53–0.85) | 0.001 | 0.76 (0.58–0.99) | 0.045 |
| Richer | 1461 | 304 | 0.46 (0.36–0.59) | < 0.001 | 0.56 (0.42–0.74) | < 0.001 |
| Richest | 1183 | 208 | 0.39 (0.29–0.53) | < 0.001 | 0.73 (0.52–1.04) | 0.078 |
| **Sex of the child** | | | | | | |
| Male | 3736 | 1267 | Ref | | Ref | |
| Female | 3702 | 1048 | 0.84 (0.73–0.96) | < 0.001 | 0.81 (0.70–0.93) | 0.004 |
| **Age of child (months)** | | | | | | |
| 0–11 | 1868 | 302 | Ref | | Ref | |
| 12–23 | 1474 | 460 | 1.93 (1.53–2.45) | < 0.001 | 1.78 (1.28–2.49) | 0.001 |
| 24–35 | 1344 | 481 | 2.22 (1.74–2.83) | < 0.001 | 2.08 (1.51–2.87) | < 0.001 |
| 36–47 | 1371 | 486 | 2.19 (1.76–2.74) | < 0.001 | 2.22 (1.60–3.07) | < 0.001 |
| 48–59 | 1381 | 585 | 2.62 (2.05–3.36) | < 0.001 | 2.97 (2.11–4.17) | < 0.001 |
| **Child anaemia status** | | | | | | |
| Sever | 142 | 119 | Ref | | Ref | |
| Moderate | 3337 | 1260 | 0.45 (0.30–0.67) | < 0.001 | 0.54 (0.35–0.85) | 0.008 |
| Not anaemic | 2822 | 766 | 0.32 (0.21–0.49) | < 0.001 | 0.33 (0.21–0.53) | < 0.001 |

(*Continued*)

**Table 5.** (Continued)

| Variables | Underweight | | Model 0 | | Model 5 | |
|---|---|---|---|---|---|---|
| | No | Yes | COR (95% CI) | p value | AOR (95% CI) | p value |
| **Size of child at birth** | | | | | | |
| Larger than average | 2485 | 565 | Ref | | Ref | |
| Average | 3243 | 930 | 1.26 (1.06–1.50) | 0.01 | 1.27 (1.06–1.52) | 0.009 |
| Smaller than average | 1711 | 818 | 2.10 (1.71–2.59) | < 0.001 | 1.96 (1.58–2.44) | < 0.001 |
| **Child's birthweight** | | | | | | |
| > = 2500 grams | 1031 | 149 | Ref | | Ref | |
| < 2500 grams | 118 | 60 | 3.52 (2.03–6.10) | < 0.001 | 2.43 (1.31–4.52) | 0.015 |
| **Diarrhoea in 2 weeks** | | | | | | |
| Yes | 824 | 361 | 1.48 (1.18–1.86) | < 0.001 | 1.61 (1.26–2.07) | 0.001 |
| No | 6600 | 1949 | Ref | | Ref | |
| **Maternal BMI (Kg/m$^2$)** | | | | | | |
| Underweight | 1205 | 526 | 3.44 (2.34–5.06) | < 0.001 | 2.51 (1.67–3.79) | < 0.001 |
| Normal | 5593 | 1703 | 2.40 (1.68–3.43) | < 0.001 | 1.85 (1.27–2.70) | 0.001 |
| Overweight | 559 | 71 | Ref | | Ref | |
| **Minimum food consumed in 24hrs** | | | | | | |
| < 4 food groups | 2267 | 661 | 1.13 (0.74–2.97) | 0.077 | 1.79 (0.79–4.09) | 0.236 |
| > = 4 food groups | 618 | 160 | Ref | | Ref | |

AOR (adjusted odds ratio); ANC (antenatal care); BMI (body mass index); COR (crude odds ratio); Ref (reference group); SNNP (southern nations, nationalities and people); WASH (water, sanitation and handwashing); Model 0, results from unadjusted univariable analysis; Model 5, adjusted for combined WASH plus all variables with p value < 0.05 in Model 1.

**Wasting.** There were 9607 children, weighted, 0–59 months of age in the analysis with WHZ data, of which 970 (10.09%) were cases of wasting. Access to improved individual water, sanitation and handwashing (S3 Table) as well as combined WASH facilities were not predictors of wasting in children, when adjusted for confounders (Table 6). The highest odds of childhood being underweight was in Somali (AOR = 5.13; 95% CI: 2.57–10.24, p < 0.001), Afar (AOR = 3.19; 95% CI: 1.61–6.32, p = 0.001) and Gambella (AOR = 2.99; 95% CI: 1.52–5.87, p = 0.002) compared with the capital city, Addis Ababa. Children from families in higher wealth quintiles were 42% less likely to be underweight compared with children from families in the lowest wealth quintile (AOR = 0.58; 95% CI: 0.39–0.87, p = 0.008). The odds of being wasted were increased by 1.05 for each additional child in preceding births (AOR = 1.05; 95% CI: 1.01–1.09, p = 0.03). The odds of wasting were 1.54 times higher (AOR = 1.54; 95% CI: 1.19–1.99, p = 0.001) among children who were a smaller size than average at birth compared with their counterparts. Children of mothers with a BMI = < 18.4 kg/m$^2$ had 2.75 times odds of wasting compared with mothers with a BMI > 24.9 kg/m$^2$ (AOR = 2.75; 95% CI: 1.36–5.58, p = 0.005) (Table 6).

## Discussion

The findings of study confirm the association between access to WASH facilities and child linear growth failure (stunting). In this study, 33% of child linear growth failure in Ethiopia may have been prevented by access to improved combined WASH facilities. This shows that nutrition-sensitive WASH interventions may be one of the major entry points alongside nutrition interventions for tackling child liner growth failure. A previous study found that improved child growth was not only a food security issue but is also closely associated with access to

**Table 6. Association between individual and combined access to WASH and wasting among children 0–59 months of age, EDHS 2016 (n = 9607).**

| Variables | Wasting | | Model 0 | | Model 5 | |
|---|---|---|---|---|---|---|
| | No | Yes | COR (95% CI) | p value | AOR (95% CI) | p value |
| **Water facility** | | | | | | |
| Improved | 4902 | 512 | 0.86 (0.68–1.08) | 0.051 | | |
| Unimproved | 3735 | 458 | Ref | | | |
| **Sanitation facility** | | | | | | |
| Improved | 848 | 87 | 0.91 (0.69–1.21) | 0.357 | | |
| Unimproved | 7789 | 882 | Ref | | | |
| **Handwashing facility** | | | | | | |
| Improved | 4604 | 491 | 0.90 (0.73–1.12) | 0.155 | | |
| Unimproved | 4033 | 479 | Ref | | | |
| **WASH facilities** | | | | | | |
| Improved | 497 | 40 | 0.72 (0.47–1.10) | 0.127 | 1.13 (0.73–1.75) | 0.588 |
| Unimproved | 8140 | 930 | Ref | | Ref | |
| **Region** | | | | | | |
| Addis Ababa | 202 | 8 | Ref | | Ref | |
| Tigray | 574 | 74 | 3.45 (2.01–5.91) | < 0.001 | 2.19 (1.16–4.12) | 0.016 |
| Afar | 76 | 17 | 5.97 (3.39–10.52) | < 0.001 | 3.19 (1.61–6.32) | 0.001 |
| Amhara | 1684 | 187 | 2.97 (1.72–5.16) | < 0.001 | 1.96 (1.00–3.83) | 0.051 |
| Oromia | 3774 | 448 | 3.18 (1.87–5.41) | < 0.001 | 2.16 (1.12–4.18) | 0.022 |
| Somali | 313 | 94 | 8.07 (4.54–14.35) | < 0.001 | 5.13 (2.57–10.24) | < 0.001 |
| Benishangul-Gumuz | 90 | 11 | 3.25 (1.79–5.90) | < 0.001 | 2.24 (1.20–4.55) | 0.027 |
| SNNP | 1853 | 123 | 1.78 (1.01–3.15) | 0.0469 | 1.29 (0.66–2.52) | 0.461 |
| Gambela | 19 | 3 | 4.35 (2.38–7.95) | < 0.001 | 2.99 (1.52–5.87) | 0.002 |
| Harari | 18 | 2 | 3.30 (1.76–6.17) | < 0.001 | 2.49 (1.23–5.06) | 0.012 |
| Dire Dawa | 33 | 4 | 3.12 (1.69–5.75) | < 0.001 | 2.13 (1.09–4.16) | 0.027 |
| **Household head** | | | | | | |
| Male | 7449 | 856 | Ref | | Ref | |
| Female | 1188 | 114 | 0.83 (0.63–1.09) | 0.073 | 0.76 (0.57–1.01) | 0.062 |
| **Wealth index** | | | | | | |
| Poorest | 1923 | 312 | Ref | | Ref | |
| Poorer | 2037 | 221 | 0.67 (0.49–0.92) | 0.012 | 0.80 (0.55–1.16) | 0.238 |
| Middle | 1809 | 210 | 0.72 (0.52–0.99) | 0.043 | 0.87 (0.61–1.24) | 0.451 |
| Richer | 1610 | 121 | 0.46 (0.32–0.67) | < 0.001 | 0.58 (0.39–0.87) | 0.008 |
| Richest | 1258 | 106 | 0.52 (0.36–0.75) | 0.001 | 0.78 (0.50–1.22) | 0.282 |
| **Childbirth order** | 8637 | 970 | 1.06 (1.02–1.10) | < 0.001 | 1.05 (1.01–1.09) | 0.030 |
| **Size of child at birth** | | | | | | |
| Larger than average | 2769 | 252 | Ref | | Ref | |
| Average | 3707 | 402 | 1.20 (0.93–1.52) | 0.151 | 1.19 (0.94–1.52) | 0.156 |
| Smaller than average | 2160 | 317 | 1.61 (1.25–2.08) | < 0.001 | 1.54 (1.19–1.99) | 0.001 |
| **Maternal BMI (Kg/m$^2$)** | | | | | | |
| Underweight | 1466 | 241 | 2.98 (1.58–5.59) | 0.001 | 2.75 (1.36–5.58) | 0.005 |
| Normal | 6499 | 684 | 1.91 (1.07–3.41) | 0.030 | 1.96 (1.03–3.76) | 0.042 |
| Overweight | 590 | 33 | Ref | | Ref | |

AOR (adjusted odds ratio); BMI (body mass index); COR (crude odds ratio); Ref (reference group); SNNP (southern nations, nationalities and people); WASH (water, sanitation and handwashing); Model 0, results from unadjusted univariable analysis; Model 5, adjusted for combined WASH facilities plus all variables with p value < 0.05 in Model 1.

WASH [28]. A systematic review and meta-analysis found that combined WASH interventions reduced the risk of child linear growth failure by 13% [29]. However, two previous studies, one from Kenya [30] and another from Bangladesh [31] found no effect of individual and combined WASH interventions on child linear growth. Although the Null et al and Luby et al studies may have had high internal validity, as is generally the case with randomized controlled [32], the external validity of these findings is less clear: in both studies, participants already had access to improved drinking water sources, had basic latrines as well as rates of open defecation at baseline. These two trials may have shown positive effects of WASH interventions on child growth if they had been conducted in areas with open defecation, inadequate handwashing, and inadequate safe water supply, as is often observed in countries in Sub-Saharan Africa and South Asia [7]. In one study it has been suggested that causal pathways linking poor WASH to child linear growth failure are complex, spanning multiple direct biological routes [33].

The current study found household access to improved combined sanitation with handwashing facilities had 29% reduced odds of stunting. This finding has important implications by identifying components of WASH which may be more focused for interventions and have synergistic effect on stunting. The current finding is consistent with a study by Humphrey [16] which showed an inverse association between linear growth and poor access to sanitation and handwashing. Some authors [16, 34] have speculated there is a link between child linear growth failure and poor hygiene as well as sanitation which may be caused by tropical enteropathy (via poor nutrient absorption) and not solely attributable to diarrhea. If there is validity to this idea, it may help to explain why the current study did not find a statistically significant association between the presence of diarrhea in the two weeks prior to the study and child linear growth failure. In a non-randomized study, Arnold et al [34] noted that linear growth failure results from bacteria exposure that is insufficient to cause symptomatic diarrhea in young children, while being sufficient to cause intestinal enteropathy.

The results of this study did not show a statistically significant effect of access to improved combined water with sanitation as well as water with handwashing facilities on child linear growth. The present findings seem to be consistent with other research which found no synergistic effect of combined water with sanitation on child linear growth failure [35, 36]. However, the findings of the current study do not support the previous research conducted by Arnold [37] and Checkley et al [38] who found a positive effect of combined water with handwashing on child linear growth. A such, there appears to be conflicting evidence of the combined effect of water with sanitation as well as combined water with handwashing on child linear growth. On the other hand, the suggestions of Checkley et al [38], that a positive relationship between improved water sources and child linear growth existed only when it was combined with improved sanitation water storage practices, provides another way of viewing this conflicting evidence.

The current findings did not show that access to improved drinking water sources reduced child linear growth failurewhich is in agreement with the previous findings [35, 39–42]. Our finding corroborates the ideas of Hunter and colleagues [43] who suggested that improved drinking water sources do not necessarily reflect the reliability or quantity of supply which could be affected by other hygiene related behaviors including household water storage and overall hygiene practices. Another possible explanation for the lack of association in our analysis is there may be a weak link between high service coverage of improved sources only [17] and child linear growth, while the role of safe water quality alone may have been undermined. This indicates that self-reported improved drinking water sources overestimate actual safe practice, which in reality may be less than optimal which could undermine the association of behavior related practices on sanitation and hygiene. The current findings also differ from a

study conducted in Eastern Ethiopia [44] that found children with access to unprotected water sources had 92% odds of being stunted compared with children with protected drinking water sources. One should be cautious in generalizing the findings of Yisak et al [45] to other parts of Ethiopia that study included a relatively small sample of children. On the other hand, analyses based on data from 171 DHS studies from 70 low- and middle-income countries found that improved drinking water sources reduced child linear growth failure by 8%.

Our study did not detect any evidence of an association between access to improved sanitation facilities and child linear growth. Although, this finding differs from findings elsewhere [35, 39, 42, 45, 46], it is consistent with that of the Burkina Faso study [41]. The current study has been unable to demonstrate a statistically significant association between access to improved handwashing (water and soap) facilities and child linear growth. This may be explained by the fact that the presence of soap and water on premises may not necessarily reflect handwashing practices at critical times such as before and after meal preparation, eating and after visiting the toilet. Biran et al in 2008 made the valid point that estimating handwashing through observation of facilities with soap may be poorly associated with actual handwashing practices [47] and even among those with access, handwashing is often inadequately practiced [48].

In this study, access to improved handwashing facilities alone was found to reduce the odds of being underweight by 17%. It may be suggested that underweight children are an indicator of chronic or acute under-nutrition which might be caused by diarrhea reflecting acute weight loss. In the current study, children with diarrhea in the two weeks prior to the study had 61% higher odds of being underweight. A study by Langford and colleagues [49] suggested handwashing may reduce severe forms of infection, but is not sufficient to reduce levels of subclinical mucosal (often chronic) damage that is strongly associated with growth faltering. The current study found that access to improved water and sanitation alone, as well as combined WASH facilities were not significantly associated with being underweight.

The present study showed that access to improved water, sanitation and handwashing as well as combined WASH facilities were not significantly associated with wasting. This is in line with findings from previous studies elsewhere [37, 41, 49]. For instance, a study conducted in Nepal showed no association between improved handwashing and wasting among Nepalese children [49]. In contrast, a previous study found increased odds of wasting among children with unprotected water sources compared with children who had access to protected sources [50].

The current study used a representative population-based data with a high response rate and conducted all analyses by adjusting for weighting, clustering and stratification to account for a complex survey design. This study has some limitations. As is often the situation in observational studies, it is difficult to quantify and even harder to rule out potential biases (recall and measurement error) in the current study. Cause-effect relationships could not be established in the present study due to the study design. Also, our study has examined access to WASH facilities rather than WASH practices. As such, the true effect of WASH practices on child growth outcomes may be underestimated as access to WASH facilities are only approximations of the actual conditions faced by children in households. In addition, the effect of microbiological, chemical and physical properties of drinking water sources on child growth outcomes is unknown due to a lack of relevant data in EDHS.

## Conclusions

After we controlled for potential confounding variables of child, mother or caregiver and household, the present study showed a protective effect of access to improved combined

WASH facilities on the risk of child linear growth failure (stunting) in Ethiopia. Access to handwashing facilities showed protective association of being underweight. The present study confirms previous suggestions and contributes additional evidence that access to water, sanitation and handwashing has a synergistic effect on child linear growth. The findings also infer the importance of not only providing WASH facilities but also educating people about how and when to properly make use of the facilities to improve health outcomes. Further research with robust methods from large randomized controlled trials would be important to examine more closely the link between the behavioral related practice of WASH and child growth outcomes. In Ethiopia, the prevalence of child growth failure is substantial, and a reasonable approach has to be developed to address major predictors associated with child growth failure in the country.

## Supporting information

**S1 Table. Univariable and multivariable analyses results of stunting for all confounding variables.**
(DOCX)

**S2 Table. Univariable and multivariable analyses results of underweight for all confounding variables.**
(DOCX)

**S3 Table. Univariable and multivariable analyses results of wasting for all confounding variables.**
(DOCX)

## Acknowledgments

The authors are grateful to The University of New South Wales (UNSW), Australia for providing PhD scholarship for the primary author. The data used in this study were taken from DHS Archive and the authors acknowledge the DHS Program for allowing us to use the dataset. The analysis and interpretation of the findings in this study are the responsibility of the authors and do not necessarily reflect the view of the DHS Program.

## Author Contributions

**Conceptualization:** Tolesa Bekele.

**Data curation:** Tolesa Bekele.

**Formal analysis:** Tolesa Bekele.

**Methodology:** Tolesa Bekele.

**Resources:** Bayzidur Rahman.

**Supervision:** Bayzidur Rahman, Patrick Rawstorne.

**Validation:** Bayzidur Rahman, Patrick Rawstorne.

**Writing – original draft:** Tolesa Bekele.

**Writing – review & editing:** Bayzidur Rahman, Patrick Rawstorne.

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
