## [Decision Letter · Decision Letter 0]

23 Mar 2020

PONE-D-19-34344

The combined effect of access to water, sanitation and handwashing on child growth indicators: Evidence from Ethiopia Demographic and Health Survey 2016

PLOS ONE

Dear Mr Bekele,

Thank you for submitting your manuscript to PLOS ONE. After careful consideration, we feel that it has merit but does not fully meet PLOS ONE’s publication criteria as it currently stands. Therefore, we invite you to submit a revised version of the manuscript that addresses the points raised during the review process.

We would appreciate receiving your revised manuscript by May 07 2020 11:59PM. To enhance the reproducibility of your results, we recommend that if applicable you deposit your laboratory protocols in protocols.io, where a protocol can be assigned its own identifier (DOI) such that it can be cited independently in the future. For instructions see: http://journals.plos.org/plosone/s/submission-guidelines#loc-laboratory-protocols

We look forward to receiving your revised manuscript.

Kind regards,

Hesham

Hesham M. Al-Mekhlafi, PhD

Academic Editor

PLOS ONE

Additional Editor Comments (if provided):

Academic Editor’s Comments:

Dear authors,

The reviewers have appreciated the work; however, they highlighted some major drawbacks that should be adequately addressed in your revised manuscript. Please consider major revision of the structure and writing of your manuscript.

1- The manuscript (all sections) should be shortened substantially and rewritten in a succinct focused manner.

2- Statistical analysis results should be revised. For instant,

- What is the standard error added to tables 1 &2?; What is the “Maternal age in years” variable in table 3; it looks incomplete!!

- Weightage description is unclear (weighted numbers should be less than actual number due to over-sampling), and numbers discrepancies are confusing. e.g we included 9607 weighted number (line 312), of which 970 (10.09%) were cases of wasting (line 313) and total of wasted in table 5 variables is 959!!!

- Totals for some variables are inconsistent (e.g. Table 5, Household head: total of wasted is 959; Combined WASH: total is 960! Maternal BMI: total is 958). This is applied to tables 4 & 5 too.

- Child anaemia status & Maternal anaemia status are variables in the tables but nothing was mentioned about anaemia assessment in methods!!

- Why different variables are displayed in tables 3-5? e.g. Child anaemia status was not included in table 5! Variables names and order should be consistent in all tables, where applicable.

Journal Requirements:

"The authors are grateful to The University of New South Wales, Australia for providing full scholarship for the primary author."

Reviewers' comments:

Reviewer's Responses to Questions

**Comments to the Author**

1. Is the manuscript technically sound, and do the data support the conclusions?

Reviewer #1: Yes

Reviewer #2: Partly

2. Has the statistical analysis been performed appropriately and rigorously? 

Reviewer #1: I Don't Know

Reviewer #2: No

3. Have the authors made all data underlying the findings in their manuscript fully available?

Reviewer #1: Yes

Reviewer #2: Yes

4. Is the manuscript presented in an intelligible fashion and written in standard English?

Reviewer #1: Yes

Reviewer #2: Yes

5. Review Comments to the Author

Reviewer #1: The study looks into effect of access to WASH facilities on child growth outcomes in Ethiopia based on second data from Ethiopia Demographic and Health Survey. The methods employed are sound. The results are largely sufficiently presented - though more figures could have been developed. Some of the results are in contrary to what is generally believed. Major revisions are required in the discussion section. Moreover, your discussion requires a structural revision. First, summarize your major findings from this study. Then you will discuss it by comparing with previous studies. Be focused. Do not confuse readers by jumping into data of previously done studies before you clearly outline your major findings. The whole objective of a discussion section is to tell your major findings and their implications. This will be corroborated by comparing your results with those findings from previous studies.

Here are my comments/edits that require revision:

1. Title - please revise the title - remove "The combined"

2. First line of the abstract [Line #16] - remove "Together with other factors, the dearth of"

3. Line 17-18: Edit the sentence as “The role of access to WASH on child growth outcomes in Ethiopia is largely unknown”

4. Line 23 - Edit as "The logistic regression model was applied"

5. Line 46 - WHO - remove "'s"

6. Please limit the number of acronyms you utilized. Some of them are not important at all. Eg; LMICs [Line 49]

7. Line 52: the acronym "(UNICEF)" should come before "Emergency Fund"

8. Line 47-48 Revise as "According to the 2019 UNICEF report, an estimated 22.2%, 7.5%, 5.6% of under-5 children were stunted, wasted, and underweight globally, respectively"

9. Line 50 - Revise "Asia and Africa are the most disproportionately affected regions"

10. Citation of Reference 5 [Line 54] requires editing. First of all, this is a UNICEF report (not a joint report). You need to correctly write the reference in the reference section (here is the reference link https://www.unicef.org/media/60806/file/SOWC-2019.pdf). So please edit Line 51-53 as "In Africa, 39% of children were stunted and 27% wasted ."

11. The first paragraph of the "Background" is packed with a lot of information. Please revise it to keep it brief and focused. Give a bit of information about the global burden and then focus on Africa since your study is in Africa.

12. Line 60: revise it as "of which half of them were in sub-Saharan Africa"

13. Lin 61-44: Delete “In addition, in 2014, nearly 1.8 billion people globally were exposed to either unimproved or faecally contaminated drinking water sources [9], while in 2015, nearly 2.4 billion or 1/3rd of the world’s population lacked access to improved sanitation facilities and 13% practiced open defecation [10].” This is unnecessary detail since you already provided global data on Line 59.

14. Throughout the “Background” section, please focus on data in Africa/Ethiopia and remove unnecessary global data that only confuses readers. Only a bit of the global scenarios is sufficient to give it a perspective – you don’t need such details.

15. Line 75: Delete “in 2008”

16. Line 94-95: Revise as “According to the 2016 Ethiopia Demographic and Health Survey,…”

17. Line 99: Remove “a” before “question”

18. Line 104: Revise as “This study seeks to determine the effect of household access…. ”

Methods

19. Line 110: Revise: We obtained data from the 2016 Ethiopia Demographic and Health Survey (reference??)”

20. Line 114: Add “of the country” after “regions”

Results

21. Once you clarified HAZ, WAZ and WHZ in the “Methods” section, it would be more understandable if you use the terms “stunted, wasted or underweight” throughout the rest of your text (and in table captions)

22. When you write 95%CI – please use “-“ between the numbers instead of “to” (e.g. 95%CI = 1.27-1.89 [Line 260])

23. Line 260-261: Please revise this statement “Children who were breastfed prior to the study had 1.55 times higher (AOR = 1.55; 95% CI: 1.27 to 1.89, p < 0.001) odds of stunting compared with those who were ever breastfed.”

Discussion

24. Delete the first sentence of your discussion. We already know the aim of the study – here we (readers) want to know your major findings.

25. Line 338: Revise as “A previous study”

26. Line 342: Delete “well designed” and add “previous”

27. Line 349: revise “some authors” as “some studies”

Line 396-398: This statement is not true and does not reflect the situation in your study area, please revise it “It seems plausible that these results may be due to the availability of soap and water on premises which may be at odds with and not” reflect safe handwashing practices.

28. Line 402-404. I don’t think this is correct – e.g. Johri et al 2019 Plos One 14(1): e0209054. showed such relationship.

29. improved drinking water sources reduced child linear growth failure.

30. You need to discuss “Limitations” of your study in your discussion section. Add one paragraph before the conclusion to discuss limitations of your study.

Reviewer #2: Dear authors,

I read carefully and thoroughly this manuscript. The topic is interesting and there is a need for these kinds of analyses connecting WASH facilities and child growth outcome in developing countries. Your manuscript can be published conditional on addressing the comments below. I presented my comments and questions by following the order of the paper.

Abstract and Introduction

Line number 31: What is the outcome variable? Or how the linear growth failure is measured? It is not clear and check it.

Line 37: What do you mean by previous suggestion? what is that suggestion? not clear for the reader.

Line 46-47: Replace the World Health Organization (WHO’s) by “ the median of the reference population” and I don't think the reference about growth standards are the best references. please check.

Line 55: A references is needed about SDG 2.2.

Line 82-83: This is not clear. please add a reference about this statement or give more explanation for it. Or move the sentence to the conclusion part /limitation.

Line 99: The study lacks reviewing some important previous literature in the topic. Including one or two papers is not enough to better understand the topic. Some assumptions in the review of previous literature, rationale and methods could also benefit to fill gap in literature.

Line 100. I am wondering if you can demonstrate the mechanism how improved access to WASH facilities related to Child growth failure (anthropometric indicators of children). This can be shown by presenting the theoretical framework and by reviewing previous literatures on the topic .

Methods

124: Mention to which DHS file you take this data? Is it the household or children's file ?

126: How many enumeration areas were selected in the first stage ? How many enumerations were there in total, urban and rural? How many were selected, rural and urban? Was the selection of the EAs random?

143: What do you mean by “exposure variable” ? What is your outcome variable? please define and describe your outcome variable in detail.

143: The authors do not show whether there is correlation between variables? At least mention if there is a problem or correlation or not.

163: Please provide the reasons for the selection of each of the five models and the reason why you considered only significant explanatory variables in each model?

163: The authors need to justify the specifications of the model. This is largely dealt with by reference to other literature using these common variables or features.

Result

191: The authors focus more on descriptive part for the variables. Your descriptive part is long. The tables are too long to understand especially table 1. You should separate the descriptive and univariate analysis and focus on the Univariate and multivariate analysis part. You can move some of the tables in the descriptive part to appendix.

209: This table is too long and difficult to understand.

210: The authors do not perform any kind of test (post estimation test).

441- 446. You wrote same title for all the tables.

6. PLOS authors have the option to publish the peer review history of their article (what does this mean?). If published, this will include your full peer review and any attached files.

Reviewer #1: No

Reviewer #2: No

---

## [Author Response · Author response to Decision Letter 0]

29 Jun 2020

June 29, 2020

Hesham M. Al-Mekhlafi

Academic Editor

PLOS ONE

Re: Manuscript ID PONE-D-19-34344

Dear Hesham M. Al-Mekhlafi:

Thank you for considering our manuscript " The effect of access to water, sanitation and handwashing on child growth indicators: Evidence from the Ethiopia Demographic and Health Survey 2016" in PLOS ONE. We have carefully addressed the editor’s and reviewers’ comments and incorporated the related changes to the main manuscript. Our responses are detailed in the enclosed Response to Reviewers document; for convenience, editor’s and reviewers’ comments are provided in black font and our responses are in blue. Thank you again for your consideration of this manuscript.

Sincerely,

Tolesa Bekele

Corresponding author

The combined effect of access to water, sanitation and handwashing on child growth indicators: Evidence from Ethiopia Demographic and Health Survey 2016

Manuscript ID PONE-D-19-34344

Response to reviewers 

Additional Editor Comments (if provided):

Academic Editor’s Comments:

1- The manuscript (all sections) should be shortened substantially and rewritten in a succinct focused manner.

Response: We have revised the main document. 

2- Statistical analysis results should be revised. For instant,

- What is the standard error added to tables 1 &2?; 

Response: We have removed this column from the tables because it shows additional information which reflects how much the sampling fluctuate a statistic will show. 

What is the “Maternal age in years” variable in table 3; it looks incomplete!!

Response: We fitted a regression model using both categorical and continuous variables. The variable “maternal age” was fitted as a continuous variable which is why this variable contains one row only in the table. We have revised the name of this variable to read as “maternal age (years)” in the main manuscript page 21, Line 276. So that readers will know it is a continuous variable.

- Weightage description is unclear (weighted numbers should be less than actual number due to over-sampling), and numbers discrepancies are confusing. e.g. we included 9607 weighted number (line 312), of which 970 (10.09%) were cases of wasting (line 313) and total of wasted in table 5 variables is 959!!!

Response: We have revised all tables to show exact number. In the Demographic and Health Survey (DHS) both under- and over-sampling were utilized by the research team, which means the weighted number is not necessarily or always less than the actual (unweighted) number. Also, during weighting there could be a rounding error based on decimal places (i.e. +1/-1). For example, as we can see from the table below from SAS output of cross-tabulation by region over wasting outcome. Similarly, it was shown in all tables in the main manuscript. 

Table of region by Wasting

Region Wasting Unweighted Frequency Weighted

Frequency Percent

Total Not wasted 7839 8637 89.9060

 Wasted 1080 969.68443 10.0940

 Total 8919 9607 

Frequency Missing = 1722

- Totals for some variables are inconsistent (e.g. Table 5, Household head: total of wasted is 959; Combined WASH: total is 960! Maternal BMI: total is 958). This is applied to tables 4 & 5 too.

Response: We have revised the tables by rerunning cross-tabulation in SAS. The reason for the variation was that there were slightly different denominators for different analyses due to variation in missing data. For example, “Maternal BMI: total is 958” which is correct as there were 1816 missing for this variable. This means we have 958/9512 in Table 6, page 29, lines 332-333. Similarly, there was missing data for some variables in Table 5. Also, total wasted was 969.68443 and not 959, which was rounded to 970.

- Child anaemia status & Maternal anaemia status are variables in the tables, but nothing was mentioned about anaemia assessment in methods!!

Response: Thanks for alerting us to this omission. The detailed DHS survey methods were given elsewhere [1] and we have provided this refence in the main document on page 7, line 132. In EDHS 2016, blood specimens for anaemia testing were collected from women aged 15-49 and all children aged 6-59 months. We have mentioned in the main document as “In the EDHS, blood samples were drawn from a drop of blood taken from a finger prick women and children (or a heel prick in the case of children age 6-59 months) and collected in a microcuvette. Haemoglobin analysis was performed on-site using a battery-operated portable HemoCue analyser.” Pages 7, lines 131-134.

- Why different variables are displayed in tables 3-5? e.g. Child anaemia status was not included in table 5! Variables names and order should be consistent in all tables, where applicable.

 Response: We accept the comment. We have three child growth outcomes in this paper: stunting, underweight and wasting. All explanatory variables were tested for association against each outcome variable. By following the model building process for each outcome, we achieved the best models that explain the outcome of interest. After performing backward elimination process, checking for potential confounders and effect modifiers we found a final model with different variables for each child growth outcome. That is, although anaemia status was included at the beginning of model building for wasting, it was not statistically significant in the final model. We have revised the tables to ensure consistency of variables names and order of variables in all tables. 

Journal Requirements:

Response: I am happy to go through the final version and edit.

Response: By the phrase “data not shown” we meant the results are not presented in a table but only in the text. We have changed the phrase accordingly on page 17, line 243 in the manuscript.

4. Thank you for stating the following in the Acknowledgments Section of your manuscript: "The authors are grateful to The University of New South Wales, Australia for providing full scholarship for the primary author." 

 Response: We have revised the information in the manuscript by removing any mention of funding of the study for which there was none. The funding statement as written is correct. UNSW is a higher educational institution that should be acknowledged for providing a PhD scholarship to the corresponding author. The authors have not received any direct funding or financial support for this particular research including study design, data collection and analysis. This project is part of PhD thesis: UNSW is the institution in which the corresponding author is enrolled as a PhD candidate. UNSW has a responsibility to monitor the progress of higher degree research candidates (e.g. corresponding author) and plays no role in the choice of study design, data collection, analysis, decision to publish or preparation of manuscript. UNSW has played no part in funding the research and nor has there been any funding for this research. Rather UNSW provides a PhD scholarship to the candidate to cover his university enrolment fees and stipend for living expenses, none of which includes funding for research. As such, the funding statement as written is correct.

Reviewers' comments:

Reviewer's Responses to Questions

Comments to the Author

1. Is the manuscript technically sound, and do the data support the conclusions?

Reviewer #1: Yes

Reviewer #2: Partly

2. Has the statistical analysis been performed appropriately and rigorously? 

Reviewer #1: I Don't Know

Reviewer #2: No

3. Have the authors made all data underlying the findings in their manuscript fully available?

Reviewer #1: Yes

Reviewer #2: Yes

 4. Is the manuscript presented in an intelligible fashion and written in standard English?

Reviewer #1: Yes

Reviewer #2: Yes

5. Review Comments to the Author

Reviewer #1: The study looks into effect of access to WASH facilities on child growth outcomes in Ethiopia based on second data from Ethiopia Demographic and Health Survey. The methods employed are sound. The results are largely sufficiently presented - though more figures could have been developed. Some of the results are in contrary to what is generally believed. Major revisions are required in the discussion section. Moreover, your discussion requires a structural revision. First, summarize your major findings from this study. Then you will discuss it by comparing with previous studies. Be focused. Do not confuse readers by jumping into data of previously done studies before you clearly outline your major findings. The whole objective of a discussion section is to tell your major findings and their implications. This will be corroborated by comparing your results with those findings from previous studies.

Response: We thank the reviewer for the comments and suggestions. We have revised the discussion part as suggested. 

Here are my comments/edits that require revision:

1. Title - please revise the title - remove "The combined"

Response: we have revised the title on page 1, line 1-2 by the removal of the word ‘combined’ which now reads: “The effect of access to water, sanitation and handwashing facilities on child growth indicators: Evidence from the Ethiopia Demographic and Health Survey 2016”

2. First line of the abstract [Line #16] - remove "Together with other factors, the dearth of"

Response: We have revised the first line of the abstract accordingly which now reads: “Poor access to water, sanitation and handwashing (WASH) facilities frequently contribute to child growth failure” on pages 1, line 16-17.

3. Line 17-18: Edit the sentence as “The role of access to WASH on child growth outcomes in Ethiopia is largely unknown”

Response: We agrees with the comment and have revised the sentence as suggested on pages 1, lines 18, by the addition of the word ‘facilities’ after the word ‘WASH’.

4. Line 23 - Edit as "The logistic regression model was applied"

Response: As advised, we have revised the manuscript accordingly on page 2, line 22. 

5. Line 46 - WHO - remove "'s"

Response: As advised, we have revised the manuscript accordingly on page 3, line 46.

6. Please limit the number of acronyms you utilized. Some of them are not important at all. Eg; LMICs [Line 49]

Response: We accept the comment and have reduced the number of acronyms throughout the paper.

7. Line 52: the acronym "(UNICEF)" should come before "Emergency Fund"

Response: We have moved the position of the acronym accordingly in the main manuscript on page 3, line 47.

8. Line 47-48 Revise as "According to the 2019 UNICEF report, an estimated 22.2%, 7.5%, 5.6% of under-5 children were stunted, wasted, and underweight globally, respectively"

Response: We have revised this part in the main document on page 3, lines 48. Now read as “According to a 2019 United Nations International Children’s Emergency Fund (UNICEF) report, an estimated 21.9% and 7.3% of under-5 children globally were stunted and wasted, respectively.”

9. Line 50 - Revise "Asia and Africa are the most disproportionately affected regions"

Response: We have edited accordingly on page 3, lines 50-51.

10. Citation of Reference 5 [Line 54] requires editing. First of all, this is a UNICEF report (not a joint report). You need to correctly write the reference in the reference section (here is the reference link https://www.unicef.org/media/60806/file/SOWC-2019.pdf). So please edit Line 51-53 as "In Africa, 39% of children were stunted and 27% wasted."

Response: We cited this refence as suggested by authors in the report: “Suggested citation: United Nations Children’s Fund (UNICEF), World Health Organization, International Bank for Reconstruction and Development/The World Bank. Levels and trends in child malnutrition: key findings of the 2019 Edition of the Joint Child Malnutrition Estimates. Geneva: World Health Organization; 2019 Licence: CC BY-NC-SA 3.0 IG.” All organizations listed as authors of the report have their own regions to cover. As such, this point estimate is the joint estimate from all regions covered by the UNICEF, WHO and World Bank. We have edited as “…whereas in Africa, 39% of children were stunted and 28% wasted” on page 3, line 53.

11. The first paragraph of the "Background" is packed with a lot of information. Please revise it to keep it brief and focused. Give a bit of information about the global burden and then focus on Africa since your study is in Africa.

Response: We accept the comment. We have revised the manuscript accordingly on pages 3-4, lines 47-69 and provided more information about the country-level burden on page 5, lines 88-95.

12. Line 60: revise it as "of which half of them were in sub-Saharan Africa"

Response: We have revised accordingly in the main document on page 4, lines 62.

13. Lin 61-44: Delete “In addition, in 2014, nearly 1.8 billion people globally were exposed to either unimproved or faecally contaminated drinking water sources [9], while in 2015, nearly 2.4 billion or 1/3rd of the world’s population lacked access to improved sanitation facilities and 13% practiced open defecation [10].” This is unnecessary detail since you already provided global data on Line 59.

Response: We have revised the sentence in the manuscript accordingly on page 3-4, lines 56-59 and reads “In 2015, an estimated 663 million of the world’s population did not have access to improved drinking water sources, of which half of them were in Sub-Saharan Africa [8]. While nearly 2.4 billion or 1/3rd of the world’s population lacked access to improved sanitation facilities and 13% practiced open defecation [9].” 

14. Throughout the “Background” section, please focus on data in Africa/Ethiopia and remove unnecessary global data that only confuses readers. Only a bit of the global scenarios is sufficient to give it a perspective – you don’t need such details.

Response: We accept the comment and we have reduced the detail referring to the global situation throughout the background. Pages 3-5, lines 40-91 in the main manuscript. 

15. Line 75: Delete “in 2008”

Response: We have revised accordingly on page 4, lines 67-69. 

16. Line 94-95: Revise as “According to the 2016 Ethiopia Demographic and Health Survey,…”

Response: We have revised as “According to the 2016 EDHS” on page 4, line 84.

17. Line 99: Remove “a” before “question”

Response: We have removed this part as it was relating to detailed global information.

18. Line 104: Revise as “This study seeks to determine the effect of household access…. ”

Response: We have edited on the manuscript accordingly on page 5, lines 93-95.

Methods

19. Line 110: Revise: We obtained data from the 2016 Ethiopia Demographic and Health Survey (reference??)”

Response: We have now included a reference and the edit appears as “Data used in this analysis were obtained from the 2016 Ethiopia Demographic and Health Survey” on page 5, lines 100-101.

20. Line 114: Add “of the country” after “regions”

Response: We have edited the manuscript accordingly on page 6, line 115.

Results

21. Once you clarified HAZ, WAZ and WHZ in the “Methods” section, it would be more understandable if you use the terms “stunted, wasted or underweight” throughout the rest of your text (and in table captions)

Response: We accept the comment and have revised the manuscript accordingly by referring to stunted, wasted or underweight in preference to the acronyms.

22. When you write 95%CI – please use “-“ between the numbers instead of “to” (e.g. 95%CI = 1.27-1.89 [Line 260])

Response: We have revised¬¬ according the manuscript. ¬¬

23. Line 260-261: Please revise this statement “Children who were breastfed prior to the study had 1.55 times higher (AOR = 1.55; 95% CI: 1.27 to 1.89, p < 0.001) odds of stunting compared with those who were ever breastfed.”

Response: We have revised the manuscript by rechecking the analysis output.

Discussion

24. Delete the first sentence of your discussion. We already know the aim of the study – here we (readers) want to know your major findings.

Response: Deleted as advised.

25. Line 338: Revise as “A previous study”

Response: We have revised the manuscript accordingly on page 30, lines 343-344.

26. Line 342: Delete “well designed” and add “previous”

Response: We have edited the manuscript accordingly on page 30, line 346.

27. Line 349: revise “some authors” as “some studies”

Response: We have edited the manuscript as advised on page 30, line 354.

Line 396-398: This statement is not true and does not reflect the situation in your study area, please revise it “It seems plausible that these results may be due to the availability of soap and water on premises which may be at odds with and not” reflect safe handwashing practices.

Response: We accept and thank you for this comment and have revised the manuscript accordingly on page 32, lines 402-405. The revised text reads as: “This result may be explained by the fact that the presence of soap and water on premises may not necessarily reflect handwashing practices at critical times such as before and after meal preparation, eating and after visiting the toilet.”

28. Line 402-404. I don’t think this is correct – e.g. Johri et al 2019 Plos One 14(1): e0209054. showed such relationship.

Response: We have edited the manuscript accordingly on page 32, Lines 402-407 and the sentence reads: “This result may be explained by the fact that the presence of soap and water on premises may not necessarily reflect handwashing practices at critical times such as before and after meal preparation, eating and after visiting toilet. Biran et al in 2008 has made the valid point that estimating handwashing through observation of facilities with soap may be poorly associated with actual handwashing practices [47] and even among those with access, handwashing is inadequately practiced [48].”

Johri et al 2019 is discussing about water, we could not find that Johri’s paper has shown the association between “handwashing and underweight.”

29. improved drinking water sources reduced child linear growth failure.

Response: We have edited accordingly in the manuscript on page 31, lines 381-382 and the sentence read: “The current study did not indicate that access to improved drinking water sources reduced child linear growth failure,….”

30. You need to discuss “Limitations” of your study in your discussion section. Add one paragraph before the conclusion to discuss limitations of your study.

Response: We agree with the comment and added a paragraph that addresses the limitations of the study on pages 33-34, lines 425-434.

Reviewer #2: Dear authors,

I read carefully and thoroughly this manuscript. The topic is interesting and there is a need for these kinds of analyses connecting WASH facilities and child growth outcome in developing countries. Your manuscript can be published conditional on addressing the comments below. I presented my comments and questions by following the order of the paper.

Abstract and Introduction

Line number 31: What is the outcome variable? Or how the linear growth failure is measured? It is not clear and check it.

Response: We have revised on page 2, lines 44-46. Child linear growth was measured by taking the length/height of a child. The outcome variables are three in this paper: stunting, underweight and wasting on page 7, lines 138-141. We have revised the manuscript which now reads: “Child growth outcome (i.e. study outcome) was shown by three indicators: stunting, underweight and wasting using the WHO 2006 Child Growth Standard [3,4]. Child linear growth failure is known as stunting (i.e. an abnormal slow rate of gain in child’s height or length) [22] which indicates chronic undernutrition in children [23].”

Line 37: What do you mean by previous suggestion? what is that suggestion? not clear for the reader.

Response: We accept the comment and have revised the manuscript on page 2, line 36, to refer to “previous findings…”

Line 46-47: Replace the World Health Organization (WHO’s) by “ the median of the reference population” and I don't think the reference about growth standards are the best references. please check.

Response: We have edited the manuscript accordingly on page 3, line 46 and replaced the refence with a more appropriate one.

Line 55: A references is needed about SDG 2.2.

Response: We have included a refence for SDG 2.2 and moved it on page 4, lines 65-66, which now read as: “WASH offers a possible solution to CGF in many countries, and its global importance is recognized in the Sustainable Development Goal (SDG) 6 while CGF is also acknowledge in SDG 2.2 [11].”

Line 82-83: This is not clear. please add a reference about this statement or give more explanation for it. Or move the sentence to the conclusion part /limitation.

Response: We have revised on page 4, lines 67-69.

Line 99: The study lacks reviewing some important previous literature in the topic. Including one or two papers is not enough to better understand the topic. Some assumptions in the review of previous literature, rationale and methods could also benefit to fill gap in literature’

Response: We have revised the manuscript by adding statement “However, the extent to which inadequate access and practices of WASH have been contributing to child growth failure remains inadequate understood. Challenges to interpret and synthesis study findings due to variations in methods used and study areas are limiting the provision of potential policy recommendations [20]. In addition, complex interaction among WASH components may also pose differential effect on child growth outcomes [9]. This study seeks to determine the role of household access to WASH facilities separately or when combined on child growth outcomes in Ethiopia. Briefly, to assess how WASH factors are associated with HAZ, WAZ and WHZ among children under-5 years old” on page 5, lines 88-95. 

Line 100. I am wondering if you can demonstrate the mechanism how improved access to WASH facilities related to Child growth failure (anthropometric indicators of children). This can be shown by presenting the theoretical framework and by reviewing previous literatures on the topic.

Response: We agree with the comment. We have reviewed what previous evidence shows about pathways between WASH and CGF on pages 3-4, lines 47-75. Details are given using conceptual frameworks in resentences on page 8, line 179 (references # 25 and 26, 27). 

Methods

124: Mention to which DHS file you take this data? Is it the household or children's file?

Response: Thanks for this suggestion. We took data from the children’s file and have now mentioned this in the manuscript on page 5, lines 100-101, which now read “Data used in this analysis were obtained from the 2016 Ethiopia Demographic and Health Survey in the children’s file [17].”

126: How many enumeration areas were selected in the first stage? How many enumerations were there in total, urban and rural? How many were selected, rural and urban? Was the selection of the EAs random?

Response: We have provided a refence for the detailed survey sampling procedures on page 7, line 135. On page 6, lines 115-123 of the main manuscript we have added: “The census frame was a complete list of 84, 915 enumeration areas (EAs) created for the 2007 PHC. An EA is a geographic area consisting of on average 181 households. In the first sampling stage, a total of 645 EAs or clusters (202 in urban and 443 in rural area) were randomly selected with probability proportional to EA size (based on the 2007 PHC) and with independent selection in each sampling stratum. In the second stage, a fixed number of 28 households per EA or cluster were selected with an equal probability systematic selection from the household list. A total of 18008 households were selected for the 2016 EDHS, of which participants from 16650 (92.5%) households were interviewed. The study participants for the current analyses included 10641 unweighted number of child-mother or caregiver pairs.”

143: What do you mean by “exposure variable?” What is your outcome variable? please define and describe your outcome variable in detail.

Response: The exposure variables are the study factors or main study variables that we have now listed on pages 7-8, lines 141-155. The outcome variables are defined in the first paragraph of the introduction page 3, lines 41-46 and include three child growth indicators: stunting, underweight and wasting as descried on page 3, lines 44-46, read: ‘A child is stunted, underweight or wasted if the z-score for height-for-age, weight-for-height, or weight-for-age, respectively falls less than -2 SDs from the World Health Organization (WHO) median of healthy refence population [3,4].’

143: The authors do not show whether there is correlation between variables? At least mention if there is a problem or correlation or not.

Response: We have revised the manuscript by providing more information on page 9, lines 183-185. The revised sentence reads: “We fitted multiple liner regression model and invoked the Variance Inflation Factor (VIF) to check for multicollinearity and there was no evidence for multicollinearity (VIF < 2).” 

163: Please provide the reasons for the selection of each of the five models and the reason why you considered only significant explanatory variables in each model?

Response: The aim of this paper is to determine the effect of WASH indicators on child growth outcomes. To assess this effect, we created five regression models containing other variables (without WASH), water variable plus significant variables retained in model one, sanitation plus model one, handwashing plus model one and combined WASH plus model one. This was just to compare the effect of each WASH indicator using these models. During the model building process, we identified explanatory variables that best explained the outcome variable (i.e. all significant variables). We retained significant variables only in each model (i.e. the best model) after testing for potential confounder variables and effect modifiers. More briefly, in model building process we used p-values for independent variables. In the logistic regression model p-values < 0.05 indicate that the term is statistically significant. We included all candidate variables with p-value < 0.25 from model 0 (maximum model) into the base model. Then using step-down procedure, we removed a single term with the highest non-significant p-value at a time until we get the model contains only significant terms (i.e. model 1). We have provided more explanation of the rationale and process on page 7-9, lines 159-185. This is the standard way of model building. Detailed information is given elsewhere [Hosmer DW, Lemeshow S. 2013. Applied logistic regression. 3rd ed. New York: Wiley. Chapter 4 and 5].

163: The authors need to justify the specifications of the model. This is largely dealt with by reference to other literature using these common variables or features.

Response: We have now provided a justification of the specifications of the model by the inclusion of the following sentence into the statistical analysis section of the manuscript on page 9, lines 176-179. The revised text reads: “We included potential confounding factors considered to be major immediate (dietary and diseases) and underlying (poverty, inadequate basic services and infrastructure) causes of child growth failure according to the conceptual frameworks of UNICEF [25] and others [26, 27].” 

Result

191: The authors focus more on descriptive part for the variables. Your descriptive part is long. The tables are too long to understand especially table 1. You should separate the descriptive and univariate analysis and focus on the Univariate and multivariate analysis part. You can move some of the tables in the descriptive part to appendix.

Response: We have split the previous Table 1 into two tables which now form Table 1 and 2 containing socio-demographic and economic characteristics of participants (Table 1) and child and maternal characteristics (Table 2) on page 13, line 219.

209: This table is too long and difficult to understand.

Response: As above, we have split the table into two bales on page 13, line 219.

210: The authors do not perform any kind of test (post estimation test).

Response: We have performed different tests in the logistic regression diagnostics such as Area under the Receiver Operating Characteristic (ROC) curve, model fit and predictive capacity of the model (i.e. on average C-statistics > 75% for all models). In addition to this we also performed linearity test for continuous variables, effect modification by fitting the multivariable regression model with interaction terms, and testing for confounders using changes in odds ratio (i.e. % change explained = [(COR-AOR)/COR)]x100%, where COR is the crude odds ratio and AOR is the adjusted odds ratio. 

441- 446. You wrote same title for all the tables.

 Response: We have now rectified this and provided unique titles for each of the tables reported on in the ‘Supporting Information’ as seen on page 35, lines 449-454. 

6. PLOS authors have the option to publish the peer review history of their article (what does this mean?). If published, this will include your full peer review and any attached files.

Do you want your identity to be public for this peer review? For information about this choice, including consent withdrawal, please see our Privacy Policy.

Reviewer #1: No

Reviewer #2: No

Reference 

1. CSA, ICF-International. Ethiopia Demographic and Health Survey 2016. Addis Ababa, Ethiopia and Rockville, Maryland, USA. 2016.

2. Hosmer DW, Lemeshow S. 2013. Applied logistic regression. 3rd ed. New York: Wiley. Chapter 4 and 5.

---

## [Decision Letter · Decision Letter 1]

4 Sep 2020

The effect of access to water, sanitation and handwashing facilities on child growth indicators: Evidence from the Ethiopia Demographic and Health Survey 2016

PONE-D-19-34344R1

Dear Dr. Bekele,

We’re pleased to inform you that your manuscript has been judged scientifically suitable for publication and will be formally accepted for publication once it meets all outstanding technical requirements.

Kind regards,

Srinivas Goli, Ph.D.

Academic Editor

PLOS ONE

Additional Editor Comments (optional):

Both reviewers feel your paper has the merit. Please address the minor comments of the reviewer 1 before I recommend it for publication.

Reviewers' comments:

Reviewer's Responses to Questions

**Comments to the Author**

1. If the authors have adequately addressed your comments raised in a previous round of review and you feel that this manuscript is now acceptable for publication, you may indicate that here to bypass the “Comments to the Author” section, enter your conflict of interest statement in the “Confidential to Editor” section, and submit your "Accept" recommendation.

Reviewer #1: All comments have been addressed

Reviewer #2: All comments have been addressed

2. Is the manuscript technically sound, and do the data support the conclusions?

Reviewer #1: Yes

Reviewer #2: Yes

3. Has the statistical analysis been performed appropriately and rigorously? 

Reviewer #1: Yes

Reviewer #2: Yes

4. Have the authors made all data underlying the findings in their manuscript fully available?

Reviewer #1: Yes

Reviewer #2: Yes

5. Is the manuscript presented in an intelligible fashion and written in standard English?

Reviewer #1: Yes

Reviewer #2: Yes

6. Review Comments to the Author

Reviewer #1: This study aims to determine individual and combined effects of access to WASH facilities on child growth outcomes in Ethiopia. The study used secondary data from a 2016 national health demography survey. This manuscript is just reporting the findings of the national survey. This is where the problem is. The study was primary led by the Central Statistic Agency of Ethiopia and its partners (https://dhsprogram.com/pubs/pdf/FR328/FR328.pdf). None of the authors are from this agency nor they showed approval to use the data for publication. In the "Ethical clearance" section, the authors said that they have received approval from the DHS program in the US - which is not the implementing agency for the survey - rather it was a funding agency. The authors need approval from the Ethiopian Central Statistical Agency for publication. What is more absurd is that the authors failed to cite the main report of the 2016 Ethiopian health survey published by the Central Statistical Agency of Ethiopia (https://dhsprogram.com/pubs/pdf/FR328/FR328.pdf) despite being the source of their data. This is unethical. Please give proper acknowledgment for the report in your Methods section.

Below find minor edits/comments:

1. Line #75: "and wash may be a solution.." edit as "suggesting that WASH may be a solution"

2. Line 94: delete "briefly" - and add "The study assesses..."

3. Please cite the source of your data source in your methods section - cite (https://dhsprogram.com/pubs/pdf/FR328/FR328.pdf)

4. Line 181: Clarify "age was not a statistical significant effect modified" - cite your data showing this (table # or figure #)

5. Line 339: Edit as "The findings of this study confirmed "

6. Line 379 "with water storage" - edit as "and water storage"

7. Line 382: delete "and this finding matches those observed in the past" - and add "which is in agreement with previous findings"

8. Line 383: edit as - Hunter and colleagues [43]

9. Line 388: Edit "It may show that" as "This indicates that"

10. Line 393: Edit as "One should be cautious in generalizing..."

11. 394: Edit as "Yishak et al [45]"

12. Line 399: delete "from some published studies" and say "from findings elsewhere"

13. Line 402: Delete "result"

14. Line 417-423: Edit as follows: "The present study showed that access to improved water, sanitation and handwashing as well as combined WASH facilities were not significantly associated with wasting. This is in line with findings from previous studies elsewhere [37, 41, 49]. For instance, a study conducted in Nepal showed no association between improved handwashing and wasting among Nepalese children [49]. In contrast, a previous study found an increased odds of wasting among children with unprotected water sources compared with children who had access to protected sources [50]."

15. Line 424: Edit as "The present study used a representative population-based data "

16. Line 426: Edit as "However, this study has some limitations"

17. Line 437: Delete "we" - Please avoid using third person pronoun throughout the text as the journal does not allow. Instead say "The present study"

18. Line 446-447: Edit as "...and a reasonable approach has to be developed to address major factors associated with child growth failure in the country. "

Reviewer #2: (No Response)

7. PLOS authors have the option to publish the peer review history of their article (what does this mean?). If published, this will include your full peer review and any attached files.

Reviewer #1: No

Reviewer #2: **Yes: **Solomon Kibret

---

## [Editor Report · Acceptance letter]

14 Sep 2020

PONE-D-19-34344R1 

The effect of access to water, sanitation and handwashing facilities on child growth indicators: Evidence from the Ethiopia Demographic and Health Survey 2016 

Dear Dr. Bekele:

I'm pleased to inform you that your manuscript has been deemed suitable for publication in PLOS ONE. Congratulations! Your manuscript is now with our production department. 

Kind regards, 

on behalf of

Dr. Srinivas Goli 

Academic Editor

PLOS ONE